# Tough, Injectable Calcium Phosphate Cement Based Composite Hydrogels to Promote Osteogenesis

**DOI:** 10.3390/gels9040302

**Published:** 2023-04-03

**Authors:** Yazhou Wang, Zhiwei Peng, Dong Zhang, Dianwen Song

**Affiliations:** 1Department of Orthopedics, Shanghai General Hospital of Nanjing Medical University, Shanghai 201600, China; 2Department of Orthopedics, Shanghai Songjiang District Central Hospital, Shanghai 201620, China; 3Department of Orthopedics, The Second Affiliated Hospital of Xuzhou Medical University, Xuzhou 221000, China; 4The Wallace H. Coulter Department of Biomedical Engineering, Georgia Institute of Technology and Emory University, Atlanta, GA 30332, USA; 5School of Medicine, Shanghai Jiaotong University, Shanghai 200240, China

**Keywords:** injectable hydrogels, osteogenesis, bone cement, biocompatible polymers

## Abstract

Osteoporosis is one of the most disabling consequences of aging, and osteoporotic fractures and a higher risk of subsequent fractures lead to substantial disability and deaths, indicating that both local fracture healing and early anti-osteoporosis therapy are of great significance. However, combining simple clinically approved materials to achieve good injection and subsequent molding and provide good mechanical support remains a challenge. To meet this challenge, bioinspired by natural bone components, we develop appropriate interactions between inorganic biological scaffolds and organic osteogenic molecules, achieving a tough hydrogel that is both firmly loaded with calcium phosphate cement (CPC) and injectable. Here, the inorganic component CPC composed of biomimetic bone composition and the organic precursor, incorporating gelatin methacryloyl (GelMA) and N-Hydroxyethyl acrylamide (HEAA), endow the system with fast polymerization and crosslinking through ultraviolet (UV) photo-initiation. The GelMA-poly (N-Hydroxyethyl acrylamide) (GelMA-PHEAA) chemical and physical network formed in situ enhances the mechanical performances and maintains the bioactive characteristics of CPC. This tough biomimetic hydrogel combined with bioactive CPC is a new promising candidate for a commercial clinical material to help patients to survive osteoporotic fracture.

## 1. Introduction

Osteoporosis, a major worldwide health problem, is associated with substantial social, economic, and public health burdens. By 2030, approximately 13.3 million individuals in the United States older than 50 years are expected to have osteoporosis [1]. Fractures, the most important consequence of osteoporosis, are associated with enormous costs and substantial morbidity and mortality [2,3]. Roughly 9 million osteoporotic fractures occur worldwide each year [4], and approximately one in three women and one in five men aged 50 years or older will have a fragility fracture during their remaining lifetime. Furthermore, a total of 23% of the subsequent fractures occur within 1 year after the first fracture, and 54.3% occur within 5 years [5], indicating the treatment for the first fracture with an internal fixation system or bone cement alone is deemed insufficient, resulting in an urgent need for early anti-osteoporosis therapy after a first fracture to prevent subsequent fractures. However, achieving rapid recovery of local fractures to avoid long-term bedrest that can lead to further systemic osteoporosis, and at the same time, improving the total bone mass to avoid secondary fractures is still a great challenge.

Osteoporosis is a systemic skeletal disease characterized by reduced bone mass and microarchitectural deterioration of bone tissue leading to an increased risk of fragility fracture [3]. Osteoporosis is a chronic disease and long-term management is required. The purpose of treating patients with osteoporosis medication is to reduce the risk of fracture and subsequent pain and disability [6]. At present, most existing therapeutics used in the treatment of osteoporosis are anti-resorptive drugs, such as bisphosphonates, and bone anabolic agents, including denosumab and teriparatide [3,6]. If individuals at high risk of fractures do not receive appropriate treatment, this may result in further consequences. Vertebral fractures are the most common among osteoporotic fractures and due to poor bone quality, screw loosening and pull-out occur frequently in older osteoporotic patients, which presents several challenges to spine surgeons [7,8].

Common strategies for improving osteointegration include aesthetic contouring at the physical level and osteogenesis at the biochemical level [9,10,11]. Percutaneous vertebroplasty (PVP) and percutaneous kyphoplasty (PKP) are widely used in the treatment of osteoporotic vertebral compression fractures (OVCFs). PVP and PKP refer to a minimally invasive spine surgery technique that injects bone cement into the vertebral body through the pedicle or beside the pedicle to relieve back pain, increase the stability of the vertebral body, and restore the height of the vertebral body [12]. Pedicle screw fixation is widely used to treat spinal disease, and the number of spine surgeries in elderly patients with osteoporosis continues to increase worldwide due to the increasingly aged population [13,14]. To improve the pull-out strength of screws in the osteoporotic spine and decrease the risk of screw loosening, several techniques are used, such as using an expandable screw, enlarging the length and diameter of the screw, and using a cement-augmented pedicle screw (CAPS). Among these approaches, CAPS has been proven to be the most effective strategy for enhancing the fixation strength to improve pedicle screw stability in patients with osteoporosis [13,14]. However, combining simple clinically approved materials to achieve good injection and subsequent molding and provide good mechanical support remains a challenge. The common types of bone cement used clinically include polymethylmethacrylate (PMMA), calcium phosphate cement (CPC), calcium sulfate cement (CSC), and composite bone cement. Currently, PMMA bone cement is the most commonly used bone cement in PVP/PKP and CAPS, having advantages such as biocompatibility, injectability, and good mechanical properties [12,15]. However, PMMA also has various disadvantages, such as it cannot be degraded, a lack of biocompatibility, a propensity to cause surrounding tissue damage due to polymerization exotherm, and residual monomer toxicity [12]. In addition, the injection of PMMA bone cement into a vertebral body increases the possibility of fracture of the adjacent vertebral body. 

The ideal bone cement is biocompatible, resorbable, osteoconductive, osteoinductive, and mechanically similar to bone. The study and development of new bone cement alternatives to PMMA is the focus of intensive investigations worldwide. In the last few decades, injectable hydrogels have gained increasing attention due to their structural similarities with the extracellular matrix, easy process conditions, and potential applications in minimally invasive surgery [15]. CPC, which has good osteoconductive and biocompatible capacity, presents an advantageous alternative material. Fortunately, with the rapid development in nanotechnology, nanomaterials are easily characterized (such as using X-ray and neutron diffraction to detect structure) [16,17,18] and evaluated (surface morphology and surface energy determined by atomic force microscopy) [19]. Thus, nanomaterials have been widely applied in the ferrimagnetic and optical domains [20,21]. The controllable preparation of CPC nano powder-formed CPC scaffolds has been widely applied in clinical application [22,23,24,25]. However, the time-consumer curing process, intrinsic uncontrolled brittleness, and poor washout resistance have limited its further integrated applications. To address these drawbacks of pure CPC, one of the major strategies is to integrate organic-inorganic phases and simulation tissue composite. Some peptide-based matrices endow materials with bioactive and mechanical properties, but they are limited by stringent synthesis processes [26,27]. Gelatin methacryloyl (GelMA) derived from collagen with injectable, bioactivity, and fast crosslinking progress has been widely studied [28,29,30,31,32]. However, its poor mechanical behavior has confined it to accelerating bone reconstruction. Introducing poly(ethylene glycol) diacrylate (PEGDA) to form GelMA/PEGDA hydrogel showed guided bone regeneration, however, the mechanical properties were still weak [28]. Thus, developing injectable, osteoblast-active, and tough materials is still a big challenge. In this study, we design an organic-inorganic precursor containing CPC, GelMA, and N-Hydroxyethyl acrylamide (HEAA) with fast gelation behavior [33,34,35,36,37]. The CPC and GelMA are supported to promote bone regeneration. The in situ-formed chemical and physical GelMA-poly(N-Hydroxyethyl acrylamide) (GelMA-PHEAA) network endows the composite with a uniquely tough structure that enhances the mechanical performance and maintains the bioactive characteristics of CPC. Due to the above properties, the resultant GelMA-PHEAA/CPC hydrogels impart superior tough mechanical properties and strong osteogenic ability.

Herein, bioinspired by natural bone structure, we develop a biomimetic bone structure that fully considers the need for appropriate interactions between inorganic osteogenic teriparatide and organic powerful biological scaffolds, achieving a scaffold that is both firmly loaded with CPC and able to provide strong mechanical support. The HEAA bridges in the system make the whole hydrogel network very tough, realizing the great storage of CPC and excellent osteogenic properties. Meanwhile, the components are Food and Drug Administration (FDA)-approved and well-suited to clinical translation. In summary, this bioactive injectable hydrogel is a novel promising therapy for fracture patients and well-suited to clinical commercialization.

## 2. Results and Discussion

### 2.1. Synthesis and Characterization of GelMA-PHEAA/CPC Hydrogels

In this study, a new injectable and bioactive hydrogel was designed. The schematic in Figure 1A illustrates the hydrogel preparation process. By introducing GelMA and HEAA monomer into the CPC precursor, a fast cross-linked homogeneous hydrogel system could be fabricated. Such a protocol allowed the in-situ gelation of hydrogels at localized defects and accelerated bone regeneration. As shown in Figure 1B, the typical vial inversion test proved that GelMA-PHEAA/CPC hydrogel could be easily formed under UV photo initiation in 2 min. The injection process in Figure 1C demonstrated that the precursor was able to plastically mold in a preset shape with fast cross-linking. Figure 1C(I–IV) show the precursor injection, defect filling, formation in situ, and final shape, respectively. These properties will allow the repairing of irregular defects in clinical applications. A rheological test was used to evaluate the processability of the used materials. Although the pre-solution contains macromolecule, monomer, and inorganic particles, the system still exhibits an obvious shear thinning phenomenon (Figure 1D). Due to the fast cross-linking ability, the system could form a stable network consisting of an organic-inorganic hybrid structure. A rheological frequency-sweep test (Figure 1E) showed that the storage modulus (G′) of the GelMA-PHEAA/CPC precursor was lower than the loss modulus (G″) at the sol stage, and after polymerization, the G′ of the GelMA-PHEAA/CPC hydrogels was always higher than the G″, which indicated that the hydrogels had both a stable structure and elasticity at a wide range of frequencies.

Figure 2A shows the SEM micrographs of the hydrogels. The pure GelMA hydrogel exhibited a smooth surface with few porous structures, while the GelMA-PHEAA hydrogel presented a flat smooth surface. However, the introduction of CPC in the GelMA system caused a remarkable increase in porosity. The CPC powder was loose within the GelMA networks. In addition, compared with the GelMA/CPC hydrogel, the GelMA-PHEAA/CPC hydrogel revealed a tighter structure, which allowed CPC to be well-dispersed in the system. The surface element detection in Figure 2B also illustrates that the GelMA-PHEAA/CPC hydrogel contains abundant bioactive particles with a homogeneous composition. The chemical structures of the precursor and hydrogels were tested using FT-IR spectroscopy (Figure 2C). The characteristic absorption peaks at 3300~3000 cm^−1^ of HEAA and the GelMA macromolecule represented the unsaturated C-H vibration. Absorption peaks at 1680~1620 cm^−1^ represent the -C=C- vibration of HEAA, which was observed in both HEAA and the GelMA-PHEAA/CPC precursor. However, in the case of GelMA, this double bond vibration may overlap with the -NH- vibration whose characteristic peak lay at 1650~1500 cm^−1^. After photo-initiation for 2 min, the double bond and unsaturated C-H vibration peaks in both GelMA-PHEAA and GelMA-PHEAA/CPC groups disappeared, indicating that both systems experienced in situ completed polymerization. In general, the complete polymerization of hydrogel precursors always results in less cytotoxicity for clinical application compared to the toxic monomers.

Next, to determine the phase composition of CPC in the composite GelMA-PHEAA/CPC hydrogels, X-ray diffraction (XRD) analysis was performed (Figure 2D). The CPC consisted of tetracalcium phosphate (TTCP, Ca_4_(PO_4_)_2_O) and dicalcium phosphate anhydrous (DCPA, CaHPO_4_), which could form hydroxyapatite (Hap) (Ca_10_(PO_4_)_6_(OH)_2_) in situ [38]. The broad peak at around 20 = 23° represented the polymer chain segment of the organic composite. Both samples exhibited typical peaks of HAp and anhydrate TTCP phase. However, the intensity of HAp and TTCP in GelMA-PHEAA/CPC was obviously weaker than CPC, which might be due to the shielding effect of the organic phase. Appropriate mechanical properties are vital for materials applied in bone defects. As mentioned, GelMA-PHEAA/CPC hydrogel could be injectable with fast cross-linking, but the form of the organic-inorganic composite needs to be strong enough to support tissue regeneration. The compress-strain curves in Figure 2E,F show a comparison of the mechanical performance of each sample. The clinically used CPC was brittle with less than 10% compressibility, which is a major disadvantage in practical application. At the same time, while, due to its bioactivity and biocompatibility, GelMA is wildly researched in tissue regeneration, the soft and weak networks it forms limited its application in hard tissue repair. The introduction of CPC in GelMA would weaken the network. As shown in Figure 2F, the modulus of GelMA/CPC is one-third that of GelMA. However, by introducing PHEAA into the system, both Young’s modulus and the compressibility of the materials were significantly promoted. GelMA/CPC and GelMA-PHEAA/CPC both showed lower mechanical properties than their hydrogel matrix. This may be attributed to the weak interaction between CPC dispersed in the system and the polymer, which decreases the strength of the polymer network connections. However, with the increase in compression deformation, the breaking stress of GelMA/CPC and GelMA-PHEAA/CPC became greater than that of their hydrogel matrix. These phenomena may be due to the strong interaction between CPC and the polymer network. In addition, the fluctuations in the curve of the GelMA-PHEAA/CPC hydrogel after 78% deformation indicated the local failure of the system; however, the materials could maintain their structural integrity at 90% strain with over 3.5 MPa stress. This property allowed the GelMA-PHEAA/CPC hydrogel to support a hard tissue structure and would reduce the potential risk of implant material rupture.

The stability of a hydrogel after swelling is very important because the mechanical property of traditional hydrogels is supposed to be weak in the swollen state. However, as shown in Figure 3A(I), the structure of GelMA-PHEAA/CPC hydrogel remained intact in the swollen state. We compressed the cylindrical hydrogel with a 500 g hook weight and it underwent deformation (Figure 3A(II,III)). After removing the force, the GelMA-PHEAA/CPC hydrogel recovered its initial shape, indicating that this hydrogel was stable in the solution environment. The swelling curve in Figure 3B exhibits the weight change of the freeze-dried hydrogels versus time. The swelling rate of pure GelMa hydrogel reached over 1000%, while, after introducing CPC or PHEAA in the system, the swelling rate of the hydrogels decreased. The PHEAA network is influenced more obviously than the inorganic component CPC. The GelMA-PHEAA/CPC hydrogel showed an appropriate ability to absorb water, and this behavior is thought to promote affinity with tissue. All hydrogels can be degraded by collagenase. The weight retention curves in Figure 3C demonstrated that the pure GelMA hydrogel degrades too fast to fill a defect, while the degradation ratio of GelMA/CPC was nearly 50% after 7 days. The hydrogel with PHEAA degraded more slowly than the single network materials, as the PHEAA promoted network density and enhanced the interaction of all components, making the system more stable. Thus, the GelMA-PHEAA/CPC needed a long time to degrade. This behavior could prolong the bioactive effect of the CPC hydrogel scaffold and slow the release of CPC, rather than rupturing quickly.

### 2.2. Cell Proliferation

As presented in Figure 4, the Live/Dead staining showed almost no dead cells among all samples after being cultured for 48 and 72 h, suggesting that the biocompatibility of GelMA and the incorporation of CPC were satisfactory, and the hydrogels do not affect the proliferation of cells on the samples. L929 cells spread and grew well on all the sample surfaces cultured for 48 h. When cultured for 72 h, all of the samples were almost covered by cells. There were plenty of living cells (green fluorescence), indicating that all hydrogels were biocompatible, which is consistent with the previous literature [39] and demonstrates that all components are well-suited for clinical translation.

### 2.3. Osteogenic Activity

The expressions of osteogenic genes including Runx2, OPN, OCN, ALP, COL I, and OSX in MC3T3 were evaluated by qRT-PCR. As shown in Figure 5A–F, the cells on the GelMA-PHEAA/CPC hydrogel sample expressed a higher level of these osteogenic-related genes than the other samples. The trends in the expressions of the four genes in cells on the four samples were consistent and ran: GelMA < GelMA-PHEAA < GelMA/CPC < GelMA-PHEAA/CPC. The increased expressions of osteogenic genes in the cells may be due to the formation of a hard tissue structure and reduced risk of CPC rupture, and the good storage of CPC, which could promote the differentiation of MC3T3 cells.

Meanwhile, the protein expression of COL I and OCN, which were analyzed using immunofluorescence staining, were further investigated to evaluate the osteogenic activity of the hydrogels. As shown in Figure 6A,B, the GelMA-PHEAA/CPC hydrogel significantly promoted COL I and OCN expression compared to the other hydrogels. Semi-quantitative statistical analysis results further confirmed that the protein expression of COL I and OCN in the GelMA-PHEAA/CPC group was notably higher than that in the GelMA/CPC and GelMA hydrogels (Figure 5C,D). These results confirmed the bioactivity of the GelMA-PHEAA/CPC hydrogel. In summary, this bioactive injectable hydrogel is a novel promising therapy for fracture patients and well-suited to clinical commercialization.

## 3. Conclusions

In this work, GelMA-PHEAA/CPC hydrogel was synthesized through a one-pot process. Due to the fast cross-linking and injectable behavior, in situ defects can be formed easily. The GelMA-PHEAA network endowed the system with a strong and tough network, and the inorganic CPC phase promotes bioactivity for bone regeneration. The GelMA-PHEAA/CPC hydrogel exhibited good biocompatibility and promoted cell proliferation. The good storage of CPC in the hydrogel system promoted the mRNA expressions of osteogenic genes (Runx2, OPN, OCN, ALP, COL I, and OSX). The improved osteogenic activity of GelMA-PHEAA/CPC was due to the increase in the CPC content and the stable hydrogel system. This study provided a reference for the modulation synthesis of injectable hydrogel, strongly supporting the contention that the biological properties of CPC can be improved by modulation synthesis to endow bone implants with good osteogenic abilities.

## 4. Materials and Methods

### 4.1. Chemicals and Reagents

Methacrylate Gelatin (GelMA), N-Hydroxyethyl acrylamide (HEAA, 98%), and Lithium Phenyl(2,4,6-trimethylbenzoyl) phosphinate (LAP, 98%) were purchased from Aladdin Reagent Inc. (Shanghai). Tetracalcium phosphate (TTCP, Ca_4_(PO_4_)_2_O) and dicalcium phosphate anhydrous (DCPA, CaHPO_4_, 98%) were supplied by Macklin Biochemical Co., Ltd. (Shanghai, China). Calcium phosphate cement (CPC) powder was prepared by equimolar mixing of TTCP and DCPA. All reagents were used as received without further purification. In this experiment, all purified water was obtained from a Millipore system with an electronic conductance of 18.2 MΩ cm.

### 4.2. Preparation of GelMA-PHEAA/CPC Hydrogels

To obtain the fast gelation bioactive hydrogel, a predetermined amount of GelMA (0.3 g) was added to 1.5 mL purified water and stirred at 60 °C to prepare a GelMA solution. Then 1.2 g HEAA, 15 mg LAP photoinitiator, and 0.3 g CPC were added into the solution and subjected to ultrasonic dispersion (see Figure 1). The obtained pre-solution was transferred into a syringe and injected into the template. After exposure to UV light (365 nm, 36 W) for 2 min, the bioactive hydrogel was fabricated and named GelMA-PHEAA/CPC. Hydrogels without CPC were defined as GelMA-PHEAA, and the 10 wt% GelMA hydrogel was chosen as a control group. The hydrogel base materials including GelMA, GelMA-PHEAA, GelMA/CPC, and GelMA-PHEAA/CPC were prepared by photo-initiation. Clinically used CPC scaffold was chosen as the control group [38].

### 4.3. Characterization of Hydrogels

Fourier-transform infrared (FTIR) spectra were acquired using a Nicolet 5700 (Thermo) at room temperature from 4000 to 400 cm^−1^. The morphology and surface elemental composition of the hydrogels were visualized under scanning electron microscopy (SEM) (3400-N, Hitachi, Tokyo, Japan). The rheological behavior of the hydrogels was evaluated by a HAAKE MARS III rheometer. The pre-solution processability was tested under rotation ramp mode from 0.01–100 s^−1^ in 1 min at 37 °C. Dynamic frequency sweep tests were carried out from 15 to 0.1 Hz at 37 °C with an oscillatory strain of 1% at the thickness of 1 mm. The microstructure of the materials was examined by X-ray diffraction (XRD, Rigaku D/Max2550, Tokyo, Japan) with a scan range of 10 to 60 degrees. The mechanical properties of hydrogels were evaluated by an electronic mechanical testing machine (SANS CMT2503, Guangzhou, China). Hydrogel samples were fabricated in a cylindrical shape (8 mm in diameter and 10 mm in height) and tested at a speed of 10 mm min^−1^. The swelling test was evaluated by gravimetric analysis. The freeze-dried hydrogel was weighed, giving W_d_, and then hydrogels were immersed in phosphate-buffered saline (PBS). The hydrogels were taken out from PBS at different time intervals and weighed again, to find W_s_, until swelling equilibrium. The swelling ratio was then calculated from swelling ratio = (W_s_ − W_d_)/W_d_ × 100%. The degradation of the samples was also recorded using gravimetric analysis. The prepared hydrogels were weighed to find W_0_ and then incubated in PBS with 2 CDU mL^−1^ collagenase type I solution at 37 °C for one week. The hydrogels were weighed every day to find W_t_. The degradation ratio was then calculated from degradation ratio = (W_0_ − W_t_)/W_0_ × 100%.

### 4.4. In Vitro Cytocompatibility Evaluation

All hydrogel specimens were immersed in sterile medium to reach a swelling equilibrium and further exposed under ultraviolet (UV) light (8 W) for another 1 h before testing. The hydrogel was soaked in fresh medium for 24 h to prepare the extracts. L929 cells were seeded in a 6-well plate for 24 h. Then, medium was replaced with extracts and incubated for 48 and 72 h. After culturing, the cells were stained with Calcein AM/PI (Servicebio, Beijing, China). Finally, the cells were viewed with fluorescence microscopy (Leica, Weztlar, Germany).

### 4.5. Quantitative Real-Time PCR (qRT-PCR) Analysis

The mRNA expressions of osteogenic genes in MC3T3-E1 cells on different samples were evaluated by using qRT-PCR. Briefly, MC3T3 cells with a cell density of 1 × 10^5^ cells/mL were seeded on samples for 7 days. At the end of the incubation time, the cells were rinsed with PBS and the total RNA was extracted with TRIzol™ reagent (Invitrogen, MA, USA). Afterward, 1.0 μg of the RNA was reverse-transcribed into complementary DNA (cDNA) by Transcript or First Strand cDNA Synthesis Kit (Roche, Switzerland). Subsequently, qRT-PCR was carried out on the Roche LightCycler480 II system using an SYBR Green I PCR Master (Roche, Switzerland). The housekeeping gene was GAPDH, and runt-related transcription factor 2 (RUNX2), osteopontin (OPN), osteocalcin (OCN), Alkaline Phosphatase (ALP), type I collagen (COL I), and Osterix (OSX) were the chosen osteogenic genes. The relative mRNA expressions of target genes were normalized to that of the reference gene GAPDH. All the primers for RT-PCR are listed in Table 1.

### 4.6. Effects of the Osteogenic Activity of MC3T3

The osteogenic activity of the hydrogels was detected by immunofluorescence staining. Briefly, immunostaining of COL I and OCN was performed after 21 days of culture at a density of 1 × 10^4^ MC3T3 cells per scaffold. After being fixed with 2.5% glutaraldehyde for 15 min, the cells were permeabilized with 0.1% Triton X-100 solution and blocked with 5% bovine serum albumin (BSA) for 1 h. Then, COL I and OCN were incubated with mouse-anti-osteocalcin IgG (Abcam, Cambridge, UK) at 4 °C overnight, followed by incubation with Alexa Fluor^®^ 647 labeled goat-anti-mouse IgG (Abcam, HK, ab150115) for 2 h. Then, F-actin was stained with phalloidin, and the nucleus was stained with DAPI (Beyotime, Shanghai, China). Subsequently, the immunofluorescence images were observed and captured by a confocal laser scanning microscopy (CLSM, A1, Nikon, Natori, Japan).

### 4.7. Statistical Analysis

All numerical data were generated by at least three separate experiments and expressed as the mean and standard deviation of each experimental condition. One-way analysis of variance (ANOVA) was used in the statistical analysis, and Tukey’s significant difference posterior test was used. Statistical significance was accepted at * *p* < 0.05, ** *p* < 0.01, and *** *p* < 0.001.

## Data Availability

The authors declare that all the data in the article are true and valid. If you need to quote from this article, please indicate the source.

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
