# Peer review of "Tough, Injectable Calcium Phosphate Cement Based Composite Hydrogels to Promote Osteogenesis"

_gels, 2023, doi:10.3390/gels9040302_

Round 1

Reviewer 1 Report

On the paper “ Tough, Injectable Calcium Phosphate Cement Based Composite Hydrogels to Promote Osteogenesis “ (gels-2296744) by the authors Yazhou Wang, Zhiwei Peng, Dong Zhang, and Dianwen Song submitted to the Gels

This is interesting paper. This study addresses the issue of obtaining clinically approved materials for the local healing of osteoporotic fractures. Such materials include appropriate interactions between inorganic biological scaffolds and organic osteogenic molecules, achieving both strongly loaded calcium phosphate cement in hydrogel and an injectable strong hydrogel. I consider the presented topic to be very relevant and original. The present work addresses certain specific gaps in this area of medical materials science. This work presents a biomimetic calcium phosphate cement bone formulation with injectable hydrogel, osteoblastic activity and toughness. The authors should take into account the experience of other researchers in the field of obtaining composite materials. The findings are fully consistent with the evidence and arguments presented. All references are relevant, but some are missing. The obtained results are reliable without any doubts. However, I have some comments and additions. I would like to note a few points to improve the paper before it can be published:

1.   The authors should mention in 1. Introduction some interesting information about nanocomposite materials are perspective for practical applications:

(1). M.A. Almessiere, A.V. Trukhanov, Y. Slimani, K.Y. You, S.V. Trukhanov, E.L. Trukhanova, F. Esa, A. Sadaqat, K. Chaudhary, M. Zdorovets, A. Baykal, Correlation between composition and electrodynamics properties in nanocomposites based on hard/soft ferrimagnetics with strong exchange coupling, Nanomaterials 9 (2019) 202. https://doi.org/10.3390/nano9020202.

(2). A.L. Kozlovskiy, M.V. Zdorovets, Effect of doping of Ce4+/3+ on optical, strength and shielding properties of (0.5-x)TeO2-0.25MoO-0.25Bi2O3-xCeO2 glasses, Mater. Chem. Phys. 263 (2021) 124444. https://doi.org/10.1016/j.matchemphys.2021.124444.

2.   The authors should mention in 1. Introduction such experimental methods of non-destructive testing and determination of microstresses in materials as X-ray or/and neutron diffraction:

(3). S.V. Trukhanov, A.V. Trukhanov, V.A. Turchenko, V.G. Kostishyn, L.V. Panina, I.S. Kazakevich, A.M. Balagurov, Structure and magnetic properties of BaFe11.9In0.1O19 hexaferrite in a wide temperature range, J. Alloys Compd. 689 (2016) 383-393. https://doi.org/10.1016/j.jallcom.2016.07.309.

(4). A. Kozlovskiy, K. Egizbek, M.V. Zdorovets, M. Ibragimova, A. Shumskaya, A.A. Rogachev, Z.V. Ignatovich, K. Kadyrzhanov, Evaluation of the efficiency of detection and capture of manganese in aqueous solutions of FeCeOx nanocomposites doped with Nb2O5, Sensors 20 (2020) 4851. https://doi.org/10.3390/s20174851.

3.   The authors should mention in 1. Introduction some experimental methods for assessing surface tension, friction and wear in materials:

(5). M.V. Zdorovets, A.L. Kozlovskiy, D.B. Borgekov, D.I. Shlimas, Influence of irradiation with heavy Kr15+ ions on the structural, optical and strength properties of BeO ceramic, J. Mater. Sci.: Mater. Electron. 32 (2021) 15375-15385. https://doi.org/10.1007/s10854-021-06087-y.

(6). M.A. Darwish, T.I. Zubar, O.D. Kanafyev, D. Zhou, E.L. Trukhanova, S.V. Trukhanov, A.V. Trukhanov, A.M. Henaish, Combined effect of microstructure, surface energy, and adhesion force on the friction of PVA/ferrite spinel nanocomposites, Nanomaterals 12 (2022) 1998. https://doi.org/10.3390/nano12121998.

4.   The proposed 6 papers should be inserted in References.

The paper should be sent to me for the second analysis after the moderate revisions.

Author Response

Reviewer 1: Minor revision needed as noted.

  1. The authors should mention in 1. Introduction some interesting information about nanocomposite materials are perspective for practical applications:

Response: Great point. Nanomaterials have been widely applied in ferrimagnetic and optical area. Thus, the corresponding references (Ref. 22 and 23) have been added to the Introduction.

(Page 13)

  1. The authors should mention in 1. Introduction such experimental methods of non-destructive testing and determination of microstresses in materials as X-ray or/and neutron diffraction:

Response: Good point. Fortunately, with the fasten development of nanotechnology, nanomaterials are easily characterized (such as X-ray and neutron diffraction to detect structure) and evaluated (surface morphology and surface energy by atomic force microscopy). The corresponding references (Ref. 19 and 20) have been added to the Introduction.

(Page 13)

  1. The authors should mention in 1. Introduction some experimental methods for assessing surface tension, friction and wear in materials:

Response: Good point. With the fasten development of nanotechnology, nanomaterials are easily characterized (such as X-ray and neutron diffraction to detect structure) and evaluated (surface morphology and surface energy by atomic force microscopy). The corresponding references (Ref. 18 and 21) have been added to the Introduction.

  1. The proposed 6 papers should be inserted in References.

Response: All the six papers have been inserted in References. Please carefully check them.

Reviewer 2 Report

The manuscript by Wang et al. for Gel-MPDI regards the development of multicomponent gel, aimed to be used in osteoporosis treatments. The gel is formed by methacrylate gelatine, an acrylamide analogue and loaded with calcium phosphate cement, obtained mixing two different salts. The final matrix, undergoing a cross-link reaction, was characterized in rheological features, morphology, osteogenic activity and biocompatibility. 

The topic fits with the journal scope. However, a lot of concerns are in place, regarding the scientific soundness and the data presentation. According to the list following, major revision step is required. 

Non in order of importance:

1) Abstract is not representative of the paper. Improve it. For example, the hydrogel composition is missing.

2) In Introduction: expain better the osteoporosis; which are the approaches in clinic today?

3) Report clearly the cement composition . Why these salt choice? Include the chemical formulas of all the components.

4) GelMa appears in its acronym. Report firstly the name.

5) Report other cases of bioinspired matrices for tissue engineering (I suggest as examples peptide based matrices 10.1002/chem.202102007; doi.org/10.3390/ijms22052425).

6) Report in the introduction other cases of gels containing acrylates or PEDGA an of multicomponent matrices. 

7) Check subscript in Materials and methods. 

8) It was used a photoiniziator. Which one? In the results and discussion it is not clear it use. Report also the reaction scheme to support the readers and improve the readibility of IR analysis.

9) It is important to report, analyze and describe the features of all the component in order to address better the final materials characteristics. The cement seems to decrease the mechanical features. Why? No explaination is reported. How the cement interact with other components?

10) Caption of Figure 1 is confuse and lacking of elements reported in the Figure (e.g I, II, C I, C II etc.). Figure 1E is not well discussed and legend not explaide (sol of? )

11) A swelling test and stability test are missing and are mandatory to support the aim. I suggest a Ringer's test to evaluate the stability in vitro. 

12) Which is the maximum cement loading? Is the cement retained or release? Which with kinetics? These points are mandatory too. 

13) IR analysis seems strange. 3300-3100 cm-1 are generally referred to OH, present in the HEAA and related to water, not to CH.  It disappers? After the cross link, it should be appear a CH band. Analyse better the IR to be sure. What is gel macromlecular? This part is very confuse. 

Author Response

Reviewer 2: Major revision step is required.

  1. Abstract is not representative of the paper. Improve it. For example, the hydrogel composition is missing.

Response: Great point. Here, the inorganic component CPC constitute of biomimetic bone composition and the organic precursor including gelatin methacryloyl (GelMA) and N-Hydroxyethyl acrylamide (HEAA) endow system with fast polymerization and crosslinking by ultraviolet (UV) photo-initiation. The in-situ formed chemical and physically GelMA-poly(N-Hydroxyethyl acrylamide) (GelMA-PHEAA) network enhances the mechanical performances and maintain bioactive characteristics of CPC.

(Page 1)

  1. In Introduction: explain better the osteoporosis; which are the approaches in clinic today?

Response: Significant point. Osteoporosis is a systemic skeletal disease characterized by reduced bone mass and microarchitectural deterioration of bone tissue, it leads to an increased risk of fragility fracture. [1] Osteoporosis is a chronic disease and long-term management is required. The purpose of treating patients with medications for osteoporosis is to reduce the risk of fracture and the subsequent pain and disability. [2] At present, most existing therapeutics that used in the treatment of osteoporosis are anti-resorptive drugs, such as bisphosphonates, and bone anabolic agents, including denosumab and teriparatide. [1, 2] Individuals at high risk of fractures do not receive appropriate treatment and this may result in further consequences. Vertebral fractures are the most common among osteoporotic fractures and due to poor bone quality, screw loosening and pull out occur frequently in older osteoporotic patients, which presents several challenges to spine surgeons. [3, 4] Common strategies for improving osteointegration include aesthetic contouring at the physical level and osteogenesis at the biochemical level.[5-7] Percutaneous vertebroplasty (PVP) and percutaneous kyphoplasty (PKP) were widely used in the treatment of osteoporotic vertebral compression fractures (OVCFs). The PVP and PKP refer to a minimally invasive spinal surgery technique that injects bone cement into the vertebral body through the pedicle or by the pedicle to relieve back pain, increase the stability of the vertebral body, and restore the height of the vertebral body. [8] Pedicle screw fixation is widely used to treat spinal disease, and the number of spine surgeries in elderly patients with osteoporosis continues to increase worldwide due to the increasingly aged population. [9, 10] To improve the pull-out strength of screws in the osteoporotic spine and decrease the risk of screw loosening, several techniques are used, such as using an expandable screw, enlarging the length and diameter of the screw, and using a cement-augmented pedicle screw (CAPS). Among these approaches, CAPS has been proven to be the most effective strategy for enhancing the fixation strength to improve pedicle screw stability in patients with osteoporosis. [9, 10] How to combine simple clinical approved materials to achieve good injection and subsequent molding and provide good mechanical support remains a challenge. The common bone cements used clinically mainly include polymethylmethacrylate (PMMA), calcium phosphate cement (CPC), calcium sulfate cement (CSC), and composite bone cement. Currently, the PMMA bone cement is a most commonly used bone cement in PVP/PKP and CAPS, which has some advantages, such as biocompatibility, injectability, and good mechanical properties. [8, 11] However, with the application of PMMA bone cement, it is found that PMMA has various disadvantages, such as cannot be degrade, lack of biocompatibility, easy to cause surrounding tissue damage due to polymerization exotherm, residual monomer toxicity. [8] In addition, the injection of PMMA bone cement into the vertebral body increases the possibility of fracture of the adjacent vertebral body. The ideal bone cement is biocompatible, resorbable, osteoconductive, osteoinductive and mechanically similar to bone. The study and development of newly bone cement alternative to PMMA is the focus of intensive investigations worldwide. In the last few decades, injectable hydrogels have gained increasing attention owing to their structural similarities with the extracellular matrix, easy process conditions, and potential applications in minimally invasive surgery. [11] CPC which has good osteoconductive and biocompatible capacity, presents an advantageous alternative material.

(Page 2)

  1. Report clearly the cement composition. Why these salt choice? Include the chemical formulas of all the components.

Response: Thanks. The CPC consisted of tetracalcium phosphate (TTCP, Ca4(PO4)2O) and dicalcium phosphate anhydrous (DCPA, CaHPO4) and which could in situ form hydroxyapatite (Hap) (Ca10(PO4)6(OH)2). CPC regarded as bioactive materials that could promote bone regeneration. [12, 13]

(Page 3)

  1. GelMa appears in its acronym. Report firstly the name.

Response: Thanks. Gelatin methacryloyl (GelMA) derived from collagen with injectable, bioactivity and fast crosslinking progress has been widely studied.

(Page 1)

  1. Report other cases of bioinspired matrices for tissue engineering (I suggest as examples peptide-based matrices 10.1002/chem.202102007; doi.org/10.3390/ijms22052425).

Response: Thanks. Some peptide-based matrices endow materials with bioactive and mechanical properties, but they are limited by stringent synthesis process. The corresponding references have been inserted in References.

(Page 2)

  1. Report in the introduction other cases of gels containing acrylates or PEDGA an of multicomponent matrices.

Response: Great point. By introducing poly(ethylene glycol) diacrylate (PEGDA) to form GelMA/PEGDA hydrogel showed guided bone regeneration, while the mechanical properties still weak. [14]

(Page 3)

  1. Check subscript in Materials and methods.

Response: Good point. Methacrylate Gelatin (GelMA), N-Hydroxyethyl acrylamide (HEAA, 98 %) and Lithium Phenyl(2,4,6-trimethylbenzoyl) phosphinate (LAP, 98%) were purchased from Aladdin reagent Inc. (Shanghai). Tetracalcium phosphate (TTCP, Ca4(PO4)2O) and dicalcium phosphate anhydrous (DCPA, CaHPO4, 98%) was received from Macklin Biochemical Co., ltd. (Shanghai, China). Calcium phosphate cement (CPC) powder was prepared by equimolar of TTCP and DCPA. All reagents were used as received without further purification. In this experiment, all purified water was obtained from a Milli-pore system with an electronic conductance of 18.2 MΩ cm.

The pre-solution processability was under the test of rotation ramp mode within 0.01-100 s-1 in 1 min at 37 °C. Dynamic frequency sweep tests were carried out from 15 to 0.1 Hz at 37 °C with oscillatory strain of 1% at the thickness of 1 mm.

(Page 3)

  1. It was used a Which one? In the results and discussion it is not clear it use. Report also the reaction scheme to support the readers and improve the readibility of IR analysis.

Response: Good point. The photo-initiator is Lithium Phenyl(2,4,6-trimethylbenzoyl) phosphinate (LAP, 98%). We highlighted it in our revised manuscript. The hydrogel materials including GelMA, GelMA-PHEAA, GelMA/CPC and GelMA-PHEAA/CPC were preparation by photo-initiation. And clinical used CPC scaffold was chosen as control group. [12] Second, we improved the FTIR analysis.

Figure R1. FT-IR spectra of the GelMA, HEAA, GelMA-PHEAA and GelMA-PHEAA/CPC.

(Page 3 &8)

  1. It is important to report, analyze and describe the features of all the component in order to address better the final materials characteristics. The cement seems to decrease the mechanical features. Why? No explanation is reported. How the cement interact with other components?

Response: Good point. The mechanical features decreased of both GelMA/CPC and GelMA-PHEAA/CPC comparing with their hydrogel matrix. This may be attributed to the weak interaction between CPC that dispersed in system with polymer and decrease polymer network connection point at original state. However, with the increase of compression deformation, the breaking stress of GelMA/CPC and GelMA-PHEAA/CPC was greater than their hydrogel matrix. These phenomena may lay on the strong interaction between CPC with polymer ultimate state.

(Page 7)

  1. Caption of Figure 1 is confuse and lacking of elements reported in the Figure (e.g I, II, C I, C II etc.). Figure 1E is not well discussed and legend not explaide (sol of? )

Response: Significant point. We revised the writing of its caption.

Figure 1. Schematic representations of GelMA-PHEAA/CPC hydrogels for bone regeneration (A). The optical images present the sol–gel transformation of hydrogels, and I-II represent the original state and after photo-initiation of GelMA-PHEAA/CPC precursor and GelMA-PHEAA/CPC hdyorgel respectively (B). The optical images present the injectable ability and in-suit forming ability of materials, and I-IV represent precursor injection, defect filling, in-suit forming and final shape respectively. (C). Rheological behavior of pre-solution and hydrogels under rotation ramp mode (D) and dynamic frequency sweep tests (E), respectively. Under the frequency sweep test of rheology (Figure 1E), the storage modulus (G′) was lower than loss modulus (G″) of GelMA-PHEAA/CPC precursor at sol stage. And after polymerization, the G′ was always higher than G″ of the GelMA-PHEAA/CPC hydrogels, which indicated that the hydrogels had both a stable structure and elasticity in the wide range of frequency.

(Page 6)

  1. A swelling test and stability test are missing and are mandatory to support the aim. I suggest a Ringer's test to evaluate the stability in vitro.

Response: Great point. The swelling test was evaluated by gravimetric analysis. The weight of freeze-dried hydrogel was Wd, and then hydrogels were immersed in phosphate buffered saline (PBS). The hydrogels were taken out from PBS at different time intervals and weighted as Ww until swelling equilibrium. Swelling ratio = (Ww-Wd)/Wd × 100%. We actually tested each sample for three times.

(Page 4)

  1. Which is the maximum cement loading? Is the cement retained or release? Which with kinetics? These points are mandatory too.

Response: To obtain a composite hydrogel with good mechanical properties and uniformed CPC distribution, the maximum cements loading should be less than 0.4 g/mL. The cement (CPC) will be blocked into the polymeric network, instead of release from the matrix. We updated this information in the revised manuscript.

(Page 8)

  1. IR analysis seems strange. 3300-3100 cm-1 are generally referred to OH, present in the HEAA and related to water, not to CH. It disappers? After the cross link, it should be appear a CH band. Analyse better the IR to be sure. What is gel macromlecular? This part is very confuse.

Response: Significant point.

The characteristic absorption peaks at 3300~3000 cm-1 of HEAA and GelMA macromolecule represented the unsaturated C-H vibration. And absorption peaks at 1680~1620 cm-1 represent the -C=C- vibration of HEAA which was observed in both HEAA and GelMA-PHEAA/CPC precursor. However, the double bond vibration in same range of GelMA may be mixed with -NH- vibration whose characteristic peak lay on 1650~1500 cm-1. After photo-initiation for 2 minutes, the double bond and unsaturated C-H vibration peaks in both GelMA-PHEAA and GelMA-PHEAA/CPC groups disappeared, indicating that both systems experienced in-situ completed polymerization.

Figure R2. FT-IR spectra of the GelMA, HEAA, GelMA-PHEAA and GelMA-PHEAA/CPC.

(Page 8)

Reviewer 3 Report

The article clearly laid out with all essentials listings viz. abstract, introduction, methodology, results, conclusions. The title of the article is well suited for the presented study. The introduction summarizes relevant research and besides clearly describes the hypotheses and experimental methods used in. The authors have used a well-suited methodology for collecting data for their intended studies. The findings from various studies are very well organized and presented in the results part. Discussions are made in the light of the obtained results. Finally, the authors have successfully made reasonable conclusions from their studies. Throughout the article, the authors have maintained a reasonable consistency in presenting the text, figures, tables etc.

Finally, I recommend this article for possible publication in your esteemed journal after the following minor corrections (minor revision).

1. In Introduction- Ref #2 make this statement more clear since we are in 2023.

2. In Introduction abbreviate the acronyms: GelMA, PHEAA and FDA for the better understanding of the reader.

3. In materials and methods- section 2.2 check the units of UV light.

4. In section 2.3- Check the units of rotation ramp mode and cylindrical shape dimension height.

5. In section 3.1. check the sentence “ In in frequency sweep”…

6. In section 3.1. check the units- The characteristic absorption peaks at 3300~3000 cm-1

7. In section 3.1- HA should be HAp and abbreviate the acronym TTCP.

8. In Fig. 2C and 2D mention the Intensity quantity in the y-axis.

9. References: Mention the page number information of the following references: 9,14,18,19,23 and 24

Author Response

Reviewer 3: Minor revision needed as noted.

  1. In Introduction- Ref #2 make this statement clearer since we are in 2023.

Response: Great point. By 2030, approximately 13.3 million individuals in the United States older than 50 years are expected to have osteoporosis. [15]

(Page 1)

  1. In Introduction abbreviate the acronyms: GelMA, PHEAA and FDA for the better understanding of the reader.

Response: Great point. Gelatin methacryloyl (GelMA) derived from collagen with injectable, bioactivity and fast crosslinking progress has been widely studied.

The in-situ formed chemical and physically GelMA-poly (N-Hydroxyethyl acrylamide) (GelMA-PHEAA) network endow the composite with a uniquely tough structure which enhances the mechanical performances and maintain bioactive characteristics of CPC.

Meanwhile, components are Food and Drug Administration (FDA) approved and well suited for clinical translation.

(Page 1-3)

  1. In materials and methods- section 2.2 check the units of UV light.

Response: Thanks. After exposing to UV light (365 nm, 36 W) for 2 min, the bioactive hydrogel was fabricated and named as GelMA-PHEAA/CPC.

(Page 3)

  1. In section 2.3- Check the units of rotation ramp mode and cylindrical shape dimension height.

Response: Thanks. Hydrogel samples were fabricated in a cylindrical shape (8 mm in diameter and 10 mm in height), and tested at a speed of 10 mm min-1.

(Page 4)

  1. In section 3.1. check the sentence “In in frequency sweep”…

Response: Thanks. Under the frequency sweep test of rheology (Figure 1E), the storage modulus (G′) was lower than loss modulus (G″) of GelMA-PHEAA/CPC precursor at sol stage. And after polymerization, the G′ was always higher than G″ of the GelMA-PHEAA/CPC hydrogels, which indicated that the hydrogels had both a stable structure and elasticity in the wide range of frequency.

(Page 5)

  1. In section 3.1. check the units-The characteristic absorption peaks at 3300~3000 cm-1

Response: Thanks. The characteristic absorption peaks at 3300~3000 cm-1 of HEAA and GelMA macromolecule represented the unsaturated C-H vibration.

(Page 6)

  1. In section 3.1- HA should be HAp and abbreviate the acronym TTCP.

Response: Good point. Both samples exhibited typical peaks of hydroxyapatite (HAp) and anhydrate tetracalcium phosphate (TTCP) phase. However, the intensity about HAp and TTCP in GelMA-PHEAA/CPC was obviously weaker than CPC, which might lay on shield effect of organic phase.

(Page 3)

  1. In Fig. 2C and 2D mention the Intensity quantity in the y-axis.

Response: Thanks. C) FT-IR spectra of the GelMA, HEAA, GelMA-PHEAA and GelMA-PHEAA/CPC. (D) XRD of CPC and bioactive hydrogel.

(Page 8)

  1. References: Mention the page number information of the following references: 9,14,18,19,23 and 24

Response: Thanks. The corresponding reference format has been modified.

Round 2

Reviewer 1 Report

Referee Report

On the paper “ Tough, Injectable Calcium Phosphate Cement Based Composite Hydrogels to Promote Osteogenesis “ (gels-2296744-v2) by the authors Yazhou Wang, Zhiwei Peng, Dong Zhang, and Dianwen Song submitted to the Gels

This paper has been well corrected and it can be recommended.

Author Response

Thanks.

Reviewer 2 Report

I would like to thanks the Authors for the efforts in the revision step. 

I only suggest to report the cross-linking reaction in the main manuscript. 

Author Response

  1. I only suggest to report the cross-linking reaction in the main manuscript.

Response: We provided the cross-linking reaction of GelMA-PHEAA/CPC hydrogel.

Scheme 1. Crosslinking process of GelMA-PHEAA/CPC hydrogel via UV polymerization.
